# A DISCRETE AND VARIATIONAL APPROACH TO SPEECH REPRESENTATION LEARNING

## ABSTRACT

Previous work on self-supervised speech representation learning shares striking similarities and calls for a unifying formalism. In this paper, we propose a flexible variational objective that generalizes recent approaches, such as VQ-APC and HuBERT, and connects to other approaches, such as VQ-CPC and wav2vec 2.0. Our main contribution is a discrete latent variable model for predictive coding, providing a framework to instantiate and connect to prior work. The learned representations of our proposed approach obtain sizable improvements on phonetic classification, speaker verification, and automatic speech recognition over prior work. Our approach provides an optimization advantage as several individually optimized terms in prior work can be optimized jointly in ours. With the coding perspective, we find that the course of training is split into distinct phases, and that it is the rate, rather than the distortion, that correlates with the downstream performance.

## 1 INTRODUCTION

Several training objectives, such as VQ-APC (Chung et al., 2020), VQ-CPC (Oord et al., 2018; van Niekerk et al., 2020), and the ones in HuBERT (Hsu et al., 2021) and wav2vec 2.0 (Baevski et al., 2020b), have been highly successful for learning speech representations. Despite the different names, these approaches share a similar goal, learning to predict one part of the input given the rest. For example, VQ-APC and VQ-CPC aim to predict the future given the past, while HuBERT and wav2vec 2.0 aim to predict the masked part of an input given the unmasked. These approaches also involve vector quantization, either as a layer in the forward process or as a separate clustering step. Most of the research has been on the differences among them, while little has been on the similarities. In this work, our goal is to provide a formalism for unifying these approaches.

Predicting one part of the input given the rest is commonly known as predictive coding, and the subject has been heavily studied in cognitive and neural science Gładziejewski (2016); Huang & Rao (2011). The premise of predictive coding is that the ability to anticipate contextual signals suggests that a model acquires correlation common across partitions of signals. In our context, the correlation could include, for instance, a phone and the acoustic frames that realize the phone, or the speaker identity and their average pitch range or their voice quality.

To adopt predictive coding as our framework for learning speech representations, we propose a graphical model with discrete latent variables. While the goal is to maximize the likelihood of the parameters[1], we derive its variational lower bound, introducing an encoder and a decoder. Depending how the input is partitioned and how the distributions are parameterized, we can instantiate and generalize the approaches in prior work. We show that k-means and cross-entropy minimization in HuBERT and WavLM (Hsu et al., 2021; Chen et al., 2022) are optimizing the terms of our lower bound. Instead of optimizing terms separately, our approach has an optimization advantage, jointly optimizing all the terms and eliminating the need of an offline clustering step. We can also instantiate contrastive objectives that maximize mutual information (Oord et al., 2018; Baevski et al., 2020b). The proposed objective is agnostic to the selection of non-overlapping contexts, providing a general framework for self-supervised representation learning.

---

[1]Technically, the probability of one part of the input given the rest is called pseudolikelihood in statistics Besag (1975). The nuance is not addressed in this paper, and we will use likelihood throughout.

Besides the theoretical motivation, there are several practical benefits when we consider a general framework. Under a single framework, it is now possible to control everything including network architectures and hyperparameters, other than the loss function. This allows us to study the difference across different approaches purely based on the change of loss functions. Some approaches require additional auxiliary losses to avoid degeneracy (Baevski et al., 2020b; Chung et al., 2021), while in our approach, terms that serve a similar purpose to these auxiliary losses naturally appear in our objectives

We pre-train Transformers with various losses on LibriSpeech, and evaluate the learned representations on phone classification, automatic speech recognition, and speaker verification. All downstream tasks are done in the probing setting, where the pre-trained Transformers are frozen. We design two scales of experiments, one with 6-layer Transformers pre-trained on the 360-hour subset of LibriSpeech and the other with 12-layer Transformers pre-trained on the entire 960 hours of LibriSpeech. The proposed losses consistently perform better than the ones being generalized.

Our framework not only learns better representations but also provides a different perspective to the learning of representations. Since our framework inherently solves a coding problem, we can inspect the rate and the distortion of compression. Specifically, the encoder takes in the input with one part missing, sends a message over to the decoder, and lets the decoder predict the missing part. The rate is measured by how much less we can transmit had the missing part is known, while the distortion is how far off the prediction of the decoder is. We find that representation learning happens in distinct phases, in-between which the distortion drops abruptly. In other words, the distortion is mostly flat during training, and it is the improvement of rate that correlates with the performance of the downstream tasks.

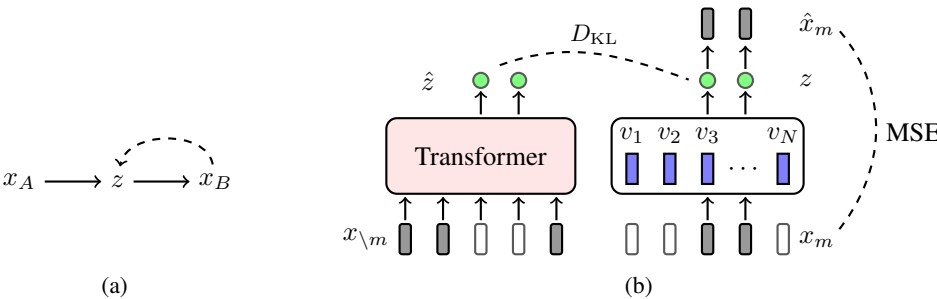

Figure 1: (a) Our proposed graphical model of a partition of the input into $x_A$ and $x_B$ with a latent variable $z$. The two solid lines denote the distributions $p(z|x_A)$ and $p(x_B|z)$, while the dashed line denotes the auxiliary distribution $q(z|x_B)$. (b) An example computation graph that realizes the graphical model. In this example, the partition consists of masked frames $x_A = x_{\setminus m}$ and unmasked frames $x_B = x_m$. There are two loss functions involved, the Kullback-Leibler divergence ($D_{\mathrm{KL}}$) measuring the rate and the mean-squared error (MSE) measuring the distortion.

## 2 A GRAPHICAL MODEL FOR PREDICTIVE CODING

We first introduce a general graphical model for predictive coding, leaving the particular choice of distributions to later sections. For any input $X$, there exists a partition of $x$ into $x_A$ and $x_B = x_{\setminus A}$. The goal of predictive coding is to predict one part given the other, and without loss of generality, we assume $x_B$ is to be predicted given $x_A$.

The two parts $x_A$ and $x_B$ is related in a generative process shown in Figure 1a, where we assume a latent variable $z$ such that $z$ only depends on $x_A$ and that knowing $z$ is sufficient to produce $x_B$. The goal is to maximize the log-likelihood $\log p(x_B|x_A)$. The log-likelihood can sometimes be intractable because the posterior $p(z|x_A, x_B)$ that encodes the information of both parts can be difficult to compute. As an alternative, an auxiliary distribution $q(z|x_A, x_B)$ is introduced to approximate the posterior $p(z|x_A, x_B)$ (Kingma & Welling, 2014; Sohn et al., 2015). We additionally assume $q(z|x_A, x_B) = q(z|x_B)$ at the expense of not being able to exactly match the posterior $p(z|x_A, x_B)$. This assumption makes sure that no distributions can access both parts of the input, adhering to the

idea of predictive coding. This leads to the variational lower bound of the log-likelihood

$$\mathcal{L}_{\text{VLB}} = D_{\text{KL}}\big(q(z|x_B)\|p(z|x_A)\big) + \mathbb{E}_{z\sim q}[-\log p(x_B|z)] \geq -\log p(x_A|x_B). \tag{1}$$

The derivation can be found in Appendix A. The distribution $p(z|x_A)$ is typically called an encoder, while the distribution $p(x_B|z)$ is typically called a decoder. In coding, the KL term is often known as the rate, and the cross entropy term is often known as the distortion. The distributions $q(z|x_B)$, $p(z|x_A)$, and $p(x_B|z)$ will have a particular parameterization in the next section.

In the context of speech processing, the input is an utterance, i.e., a sequence of $T$ frames $x = (x_1, \ldots, x_T)$, with $x_t$ for $t = 1, \ldots, T$ typically being log Mel features. We next show how we can instantiate two commonly seen settings for pre-training: future prediction and masked prediction. The differences between the two lie in how we partition the input $x_1, \ldots, x_T$ and how the prediction factorized.

## 2.1 FUTURE PREDICTION

To instantiate future prediction, we arbitrary choose a time point $t$, and partition the input sequence into $x_A = x_{<t}$ and $x_B = x_{\geq t}$. We call $x_{<t}$ the past and $x_{\geq t}$ the future. The goal is to predict the future $x_{\geq t}$ given the past $x_{<t}$. The variational bound becomes

$$\mathcal{L}_{\text{AR}} = \sum_{i=0}^{T-t} \big[ D_{\text{KL}}\big(q(z_{t+i}|x_{t+i})\|p(z_{t+i}|x_{1:t-1+i-k})\big) + \mathbb{E}_{z_{t+i}\sim q}[-\log p(x_{t+i}|z_{t+i})]\big] \tag{2}$$

$$\geq \sum_{i=0}^{T-t} -\log p(x_{t+i}|x_{1:t-1+i-k}) = -\log p(x_{t:T}|x_{1:t-1-k}) \tag{3}$$

where we make two assumptions. The first is that instead of depending on the entire future frames, the auxiliary distribution $q$ only depends on a single future frame. This choice is important as we will see in later sections. The second assumption is about the time shift $k$ used in Chung et al. (2019; 2020); Yang et al. (2022). The original motivation is to prevent models from taking advantage of temporal smoothness when making a prediction. Here, we treat it as an additional independence assumption that $p(x_{t:T}|x_{1:t-1}) = p(x_{t:T}|x_{1:t-1-k})$ for some $k \geq 0$. The distribution $p(z_{t+i}|x_{1:t-1+i-k})$ is usually parameterized with an autoregressive network, e.g., a unidirectional LSTM, or a Transformer with causal attention. In practice, $t$ is typically small so that most of the frames get predicted.

## 2.2 MASKED PREDICTION

Instead of choosing a time point for partitioning as in future prediction, the frames are not necessarily required to be contiguous in masked prediction. Formally, we choose a subset of indices $m \subseteq \{1, \ldots, T\}$ to form the partition $x_m = \{x_i\}_{i\in m}$ and $x_{\backslash m} = \{x_i\}_{i\notin m}$. Typically, the subset of indices $m$ is called a mask, and the goal is to predict the masked frames $x_m$ given the unmasked frames $x_{\backslash m}$. The variational bound in this case is

$$\mathcal{L}_{\text{MLM}} = \sum_{i\in m} \big[ D_{\text{KL}}\big(q(z_i|x_i)\|p(z_i|x_{\backslash m})\big) + \mathbb{E}_{z_i\sim q}[-\log p(x_i|z_i)]\big] \tag{4}$$

$$\geq \sum_{i\in m} -\log p(x_i|x_{\backslash m}) = -\log p(x_m|x_{\backslash m}), \tag{5}$$

where we make two assumptions. The first is again that the auxiliary distribution $q$ only depends the current frame. The second is that the masked frames are independent of each other given the unmasked ones. The distribution $p(z_i|x_{\backslash m})$ is usually parameterized with Transformer encoders (Devlin et al., 2019).

## 3 PARAMETERIZATION

In this section, we detail the exact parameterization of the distributions. We use masked prediction as an example and expand the KL term in Equation 5 to get

$$\mathcal{L}_{\text{MLM}} = \sum_{i\in m} \mathbb{E}_{z_i\sim q}\big[ \log q(z_i|x_i) - \log p(z_i|x_{\backslash m}) - \log p(x_i|z_i)\big]. \tag{6}$$

The three terms correspond to the entropy, cross entropy and the reconstruction loss, respectively.

We parameterize the prior distributions $p(z_i|x_{\backslash m})$ with a Transformer encoder as shown in Figure 1b. In masked prediction, given a sequence of output vectors $c_1, \ldots, c_T$ from the last Transformer layer, we apply a linear projection $U$ to obtain the probability of the latent variable,

$$p(z_i|x_{\backslash m}) = \frac{\exp\left(c_t^\top U \mathbf{1}_{z_i}\right)}{\sum_{j=1}^N \exp\left(c_t^\top u_j\right)}, \tag{7}$$

where $i \in m$, $\mathbf{1}_{z_i}$ is a one-hot vector with the $z_i$-th element set to 1, and $u_j$ is the $j$-th row of $U$.

We parameterize the auxiliary distribution $q(z_i|x_i)$ with a codebook $V$ as

$$q(z_i|x_i) = \frac{\exp\left(-\|x_i - V\mathbf{1}_{z_i}\|^2\right)}{\sum_{j=1}^N \exp\left(-\|x_i - v_j\|^2\right)}, \tag{8}$$

where $v_j$ is the $j$-th row of a matrix $V$ (also known as the $j$-th codeword). The distribution $q$ is defined so as to find the closest entry of $V$ to a masked frame $x_i$ in Euclidean distance. We assume the codebook has the same dimension as the input acoustic frames $x_i$.

Finally, we use the same codebook $V$ to parameterize the reconstruction distribution as follows:

$$p(x_i|z_i) = \frac{1}{(2\pi)^{d/2}} \exp\left(-\frac{1}{2}\|x_i - V\mathbf{1}_{z_i}\|^2\right), \tag{9}$$

where $d$ is the dimension of $x_i$. In other words, the reconstruction distribution uses the quantized frames $z_i$ to generate $x_i$.

## 4 RELATED WORK

Predictive coding has been the inspiration for many self-supervised approaches in speech representation learning since the work of Oord et al. (2018) and Chung et al. (2019), with both approaches having explicit connections to predictive coding. The generalization to masked prediction, such as wav2vec 2.0 (Baevski et al., 2020b) and HuBERT (Hsu et al., 2021), is heavily influenced by the advance of pre-training in text (Devlin et al., 2019). However, unlike text, speech is continuous, and to apply the same approaches in text, speech frames are quantized before pre-training; hence the vector quantization in wav2vec 2.0 and k-means in HuBERT.

The connection to predictive coding has been lost along the development of speech representation learning. There is a parallel development in computer vision. In fact, masked prediction, or inpainting, is one of the early approaches to self-supervised learning (Pathak et al., 2016), and its effectiveness has been reaffirmed in Transformers (He et al., 2022). The connection to predictive coding is also weak in that line of research. However, masked prediction is clearly used in the modeling of visual cortex (Huang & Rao, 2011).

In this section, we show how the proposed variational bound connects to several recent approaches. We will use masked prediction as an example, but the discussion equally applies to future prediction.

### 4.1 VQ-APC

VQ-APC (Chung et al., 2020) is a discrete version of Autoregressive Predictive Coding (APC) (Chung et al., 2019; Yang et al., 2022) that predicts a future acoustic frame given past frames. The generative process can be summarized with the solid lines in Figure 1a. This observation is made in Yeh & Tang (2022). They further generalize the loss function by maximizing a conditional mutual information, and arrive at the same objective as our loss in the case of future prediction.

### 4.2 HUBERT AND WAVLM

HuBERT (Hsu et al., 2021) and WavLM (Chen et al., 2022) are derivatives of DeepCluster (Caron et al., 2018), where k-means is applied to acoustic frames, and a Transformer encoder is used to predict cluster assignments of masked frames. The HuBERT loss function is defined as

$$\mathcal{L}_{\text{HuBERT}} = \sum_{i \in m} \mathbb{E}_{z_i \sim q}\left[\log p(z_i|x_{\backslash m})\right], \tag{10}$$

where $q$ is a point-mass distribution on the cluster assignment of k-means and $z_i$ is the assignment.

It is now straightforward to see that the HuBERT loss function is an instance of Equation 6. First, the k-means step optimizes $\mathbb{E}_{z_i \sim q}\big[\log p(x_i|z_i)\big]$ with the codebook $V$ being the centroids of k-means. Next, HuBERT is trained to predict hard assignments drawn from $q$ with cross entropy, i.e., the second term in Equation 6. Finally, the entropy term $\mathbb{E}_{z_i \sim q}[\log q(z_i|x_i)]$ remains zero because of $q$ being a point-mass distribution.

The codebook $V$ does not necessarily need to be updated in they are properly initialized as discovered in Chiu et al. (2022). The approach, known as BEST-RQ, is a simplification of HuBERT, and can also be instantiated in our framework. Because the codebook is randomly initialized and never updated, Best-RQ does not optimize the reconstruction term.

## 4.3 VQ-CPC AND WAV2VEC 2.0

VQ-CPC (van Niekerk et al., 2020) and wav2vec 2.0 (Baevski et al., 2020b) are vector quantized variants of contrastive predictive coding (Oord et al., 2018), with the former in the form of future prediction and the latter in the form of masked prediction. We focus on the connection between wav2vec 2.0 and the proposed masked prediction objective, but the argument equally applies to VQ-CPC.

Recall that $c_1, \ldots, c_T$ is the output of the Transformer encoder. The loss function of wav2vec 2.0 is defined as

$$\mathcal{L}_{\text{w2v2}} = \sum_{i \in m} \mathbb{E}_{z_i \sim q}\left[ -\log \frac{\exp\big(f(V\mathbf{1}_{z_i}, c_i^\top W)/\kappa\big)}{\sum_{v' \in \mathcal{D}} \exp\big(f(v', c_i^\top W)/\kappa\big)} \right], \tag{11}$$

where $f(\cdot, \cdot)$ is chosen to be the cosine similarity, $W$ projects $c_i$ from model dimension to codebook dimension, $\kappa$ is the temperature, and $\mathcal{D}$ contains the positive sample $v_i$ and other negative samples. The choice of $q$ in this case is a distribution over vector quantized output of the convolutional network in wav2vec 2.0. In practice, the expectation is approximated with Gumbel-Softmax (Jang et al., 2017).

We first note that the loss in wav2vec 2.0 is not the same as the cross entropy, unless additional assumptions are made on how negative samples $\mathcal{D}$ are drawn. The loss in wav2vec 2.0 is known as InfoNCE (Oord et al., 2018), and its connection to cross entropy is noted in Kong et al. (2019). The two are equavalent if $\mathcal{D}$ uniquely includes all code types in $V$.

We then note that there is no reconstruction term in wav2vec 2.0. However, the reconstruction can be incorporated into contrastive learning, leading to another approach in the next section. Similar to the proposed approach, the codebook and the encoder are jointly optimized in wav2vec 2.0.

In wav2vec 2.0, there is an additional auxiliary loss called the diversity loss, aiming to diversify the codebook usage. The diversity loss takes the form of an entropy, and is reminiscent to our entropy term. Our entropy term also has the effect of diversifying codebook usage. However, the diversity loss is computed on a mini-match, while our entropy is frame-based. Since the expectation over $q$ is often approximated with Gumbel-Softmax in practice, the entropy term $\mathbb{E}_{z_i \sim q}[-\log q(z_i|x_i)]$ becomes zero.

## 4.4 PROTOTYPICAL CONTRASTIVE LEARNING

Finally, Prototypical Contrastive Learning (Li et al., 2021) incorporates a clustering objective into contrastive learning for visual representation learning. The loss function under our masked prediction framework is defined as:

$$\mathcal{L}_{\text{ProtoNCE}} = \sum_{i \in m} \mathbb{E}_{z_i \sim q}\left[ -\log \frac{\exp\big(f(V\mathbf{1}_{z_i}, c_i^\top W)/\kappa\big)}{\sum_{v' \in \mathcal{D}} \exp\big(f(v', c_i^\top W)/\kappa\big)} - \log p(x_i|z_i) \right] \tag{12}$$

where the first term is the InfoNCE loss, the second term is the k-means loss with $V$ being the centroids of k-means, and $z_i$ is the assigned cluster. These two terms can be referred to as the cross entropy and reconstruction in Equation 6. Again, the entropy is zero as $q$ is a point-mass distribution.

## 5 EXPERIMENTS

We perform several experiments and compare to recent self-supervised pre-training objectives. We pre-train Transformers in SMALL and BASE configurations, where SMALL consists of 6-layer Transformer with 4 attention heads, while BASE consists of 12-layer Transformer with 8 attention heads. After pre-training, we evaluate learned representations on phone classification, speaker verification and automatic speech recognition, to see how well the representations encode phonetic and speaker information.

We pre-train SMALL models on the 360-hour subset of LibriSpeech and BASE models on the 960-hour subset of LibriSpeech (Panayotov et al., 2015). We perform downstream tasks on Wall Street Journal (WSJ) (Paul & Baker, 1992). We extract log Mel features of 40 dimensions with a 10 ms frame shift, and concatenate every two frames, leading to 80-dimensional features in a 20 ms frame shift.

We use force alignments as phonetic labels for phone classification following the protocol of (Chung et al., 2019; Yang et al., 2022). We report error rates on `dev93` and `eval92`, using 10% of the training set `si284` for development. Training targets for HuBERT are extracted with k-means of 100 centroids. The centroids are initialised with k-means++ (Arthur & Vassilvitskii, 2007). We use VoxCeleb1 for speaker verification task (Nagrani et al., 2020).

### 5.1 PRE-TRAINING

We pre-train Transformer models one with future prediction and another with masked prediction, and compare them to our HuBERT and wav2vec 2.0 under the same model configurations. We set the codebook size to 100 for all models.

Our wav2vec 2.0 and HuBERT differ from the original Baevski et al. (2020b); Hsu et al. (2021) by a few architecture simplifications. Similar to Lin et al. (2022); Misra et al. (2021), we remove the convolution layers and take Mel spectrograms as input instead of waveforms. We employ a single codebook in our wav2vec 2.0 for quantization. Rather than Post-LN Transformers, we use Pre-LN Transformers for pre-training and remove the warm-up stage (Geiping & Goldstein, 2023; Xiong et al., 2020). We simply use sinusoidal positional embeddings rather than relative position embeddings used in the original HuBERT and wav2vec 2.0. We do not use diversity loss Baevski et al. (2020b), since it does not correspond to any term in the proposed loss function. More importantly, our HuBERT does not have multiple iterations in pre-training Hsu et al. (2021) to refine targets. The refinement step involves choosing a layer with forced alignments, technically not qualified as unsupervised. While it is possible to instantiate HuBERT loss functions for multiple iterations, The goal of the paper is to show that the proposed approach instantiates HuBERT.

We train SMALL models on a single RTX 2080TI with a batch size of 8 for 100 epochs, and BASE models on a single A40 with a batch size of 16 for 150 epochs. We set a maximum length of 1400 frames per utterance, corresponding to about 28 seconds. The learning rate is fixed to $10^{-4}$ under the Adam optimizer. For masked prediction, we use a masking span of 4 frames (80 ms) with a probability of 0.2 for each frame. We choose a time shift $k$ of 2 (40 ms) for future prediction. A detailed list of hyperparameters is in Appendix B.

### 5.2 DOWNSTREAM TASKS

To study the representations learned from the proposed variational objectives, we conduct the following downstream tasks. We use MLM-VLB and Causal-VLB to refer to VLB on masked prediction (Equation 5) and future prediction (Equation 3) respectively.

**Phone Classification** We use a linear layer to examine the accessibility of phonetic information in the learned representations, i.e., the different layers of a pre-trained model. We freeze the pre-trained model and only train a linear layer with a learning rate of $10^{-3}$ for 10 epochs. We report the performance based on the layer that gives the best phone error rates (PERs) of that model.

Table 1 reports the results of phone classification. The proposed VLB on masked prediction pre-training consistently outperforms other self-supervised loss functions in terms of PERs, indicating the phonetic prominence of learned representation. The improvements are more pronounced in the

Table 1: Summary of results on phone classification using phone error rates (PERs) and speaker verification using equal error rates (EERs). In phone classification, a linear layer is trained on extracted representations, while in speaker verification, two linear layers are trained. The number of parameters (Param) and the amount of pre-trained data (Pre-train) are reported. The numbers are not comparable to SUPERB (wen Yang et al., 2021), since they rely on x-vectors (Desplanques et al., 2020).

|  | Param | Pre-train | PER(%) ↓ | | EER(%) ↓ |
|---|---|---|---|---|---|
|  |  |  | dev93 | eval92 |  |
| log Mel | - | - | 49.8 | 50.0 | 24.6 |
| i-vector | - | - | - | - | 15.7 |
| SMALL |  |  |  |  |  |
| wav2vec 2.0 |  |  | 15.9 | 15.4 | 20.8 |
| HuBERT | 42.6M | LS360 | 16.0 | 16.2 | 18.5 |
| MLM-VLB |  |  | **15.0** | **15.4** | 17.8 |
| Causal-VLB |  |  | 18.5 | 18.8 | **15.6** |
| BASE |  |  |  |  |  |
| wav2vec 2.0 |  |  | 14.2 | 12.3 | 20.2 |
| HuBERT | 85.2M | LS960 | 14.4 | 12.7 | 18.3 |
| MLM-VLB |  |  | **11.9** | **12.0** | 14.4 |
| Causal-VLB |  |  | 17.6 | 15.9 | **13.6** |

BASE setting. Not surprisingly, the performance of VLB on future prediction is worse than masked prediction due to the use of causal attention. In addition to the proposed approach, wav2vec 2.0 generally obtains better PERs than HuBERT.

**Speaker Verification**   Speech signals also convey speaker-dependent information, such as vocal tracts, accents, and speaking styles. To validate if the learned representations capture speaker characteristics, we experiment with speaker verification. Speaker verification is a task determining whether an unknown voice matches a known voice of a speaker.

To extract speaker information from learned representations, we train a speaker classifier on the training set of VoxCeleb1. We average frame representations to obtain utterance-level representations, and employ only two linear layers to predict 1251 speakers following (Fan et al., 2020). The linear classifier is optimized with a learning rate of $10^{-3}$ for 10 epochs. After training, we take the output of the first linear layer as speaker vectors for speaker verification. We use the cosine of vectors as the similarity score, computing the equal error rates (EERs) by sweeping the threshold. We set the dimension of speaker vectors to 512.

Table 1 summarizes the results on speaker verification. Our i-vector with cosine similarity achieves 15.7% (Dehak et al., 2010). In comparison to HuBERT and wav2vec 2.0, VLB on both pre-training tasks achieves lower EERs and outperforms the i-vector baseline. In particular, on future prediction, VLB reduces EERs by 5.2% and 6.6% absolute for SMALL and BASE. Unlike phone classification, speaker verification demonstrates superior performance with the suggested objective on future prediction as opposed to masked prediction.

**Automatic Speech Recognition**   We next showcase the usefulness of the learned representations with automatic speech recognition (ASR). We freeze the pre-trained model and extract the layer that corresponds to the best phone classification results for ASR. We train a sequence-to-sequence (s2s) model with characters on WSJ (Kim et al., 2017; Yang et al., 2022). The encoder contains two convolutional layers with $(32, 32)$ channels and $(2, 1)$ strides, and a 4-layer, 256-dimensional bidirectional GRU. The decoder is a unidirectional 256-dimension GRU. For inference, we use beam search with a beam size of 5. No language model is applied during inference. More training details for the s2s model are in Appendix C.

As shown in Table 2, although we use a small sequence-to-sequence model, our baseline based on log Mel features is on par with wav2letter++ (Pratap et al., 2019) on WSJ. We obtain results by

Table 2: The proposed loss functions improve WSJ performance on both testing sets in terms of CER and WER. The improvements are observed in both training configurations.

| | Param | Pre-train | dev93 | | eval92 | |
|---|---|---|---|---|---|---|
| | | | CER | WER | CER | WER |
| Baseline (log Mel) | - | - | 6.8 | 18.2 | 5.1 | 14.7 |
| wav2letter++ (log Mel)[†] | - | - | 6.3 | 19.5 | 4.1 | 13.9 |
| SMALL | | | | | | |
| wav2vec 2.0 | | | 6.7 | 18.6 | 5.4 | **14.8** |
| HuBERT | | | 6.5 | 17.6 | **5.2** | 15.2 |
| MLM-VLB | 42.6M | LS360 | **6.2** | **17.4** | 5.3 | 15.0 |
| Causal-VLB | | | 6.5 | 17.6 | 5.4 | 15.0 |
| BASE | | | | | | |
| wav2vec 2.0 | | | 5.0 | 14.5 | 4.9 | 14.4 |
| HuBERT | | | 5.2 | 15.2 | 5.0 | 14.5 |
| MLM-VLB | 85.2M | LS960 | **4.4** | **13.6** | **3.6** | **11.4** |
| Causal-VLB | | | 5.4 | 15.5 | 4.0 | 12.4 |

[†] Numbers taken from Baevski et al. (2020a).

replacing log Mel features with representations extracted from the pre-trained models. In line with the results on phone classification, VLB representations show superior word error rates (WERs) and character error rates (CERs) than other approaches, even for future prediction. In SMALL, the proposed approach obtains a 1.2% absolute improvement on dev93. The improvements are more significant in BASE, where VLB with masked prediction improves WERs by 1.6% absolute on dev93 and 3.1% absolute on eval92. VLB with future prediction also reduces WER for 2.1% absolute on eval92. Interestingly, wav2vec 2.0 obtains better results than HuBERT for most cases. While HuBERT and MLM-VLB are optimizing the same loss function, optimizing MLM-VLB directly results in superior PERs compared to separately optimizing the reconstruction loss and the KL loss with k-means and cross entropy. This eliminates the need for an extra clustering step required in HuBERT. The results from the downstream tasks demonstrate the efficacy of the proposed approach in learning representations that reveal prominent phonetic and speaker properties.

## 5.3 LEARNING DYNAMICS

In addition to downstream tasks, we analyze the learning dynamics of learning representations through the proposed VLB. Inspired by previous work Alemi et al. (2018); Prokhorov et al. (2019) that has emphasized the important roles of the KL term (rate) and the reconstruction term (distortion), we inspect the learning dynamics based on the two terms rather than the VLB alone. Specially, the rate term signifies the additional bits required to encode the prior distribution $p(z|x_A)$ using the variational distribution $p(z|x_B)$. This is of particular interest because it involves the prior distribution that is modeled by a Transformer encoder.

We describe the learning dynamics of the proposed approach via the rate and the distortion as shown in Figure 2. The learning process can be divided into several stages as indicated by the dashed lines. At the beginning of each stage, more latent codes are learned to encode the contexts, resulting in a sudden rise in the number of bits and a reduction in distortion. Once a new mode is discovered, the variational distribution remains stable with a fixed distortion until the next stage is reached, while the encoder $p(z|x_A)$ learns to minimize the extra bits needed to encode $x_A$. After around the 30th epoch, the model settles into a fixed mode in the defined contexts. In contrast to variational learning, HuBERT sees no interaction between $p(z|x_A)$ and $q(z|x_B)$ in Figure 2, since distributions on different disjoint contexts are not optimized jointly. More interestingly, representations achieve better PERs when fewer bits (lower KL loss) are needed in Figure 2.

To analyze the impact of the rate on the quality of learned representations, we consider VLB training with a codebook initialized with k-means++, denoted as MLM-VLB (k-means++). As reported in Table 3, the variant obtains a lower distortion at the expense of the rate. However, the WER degrades

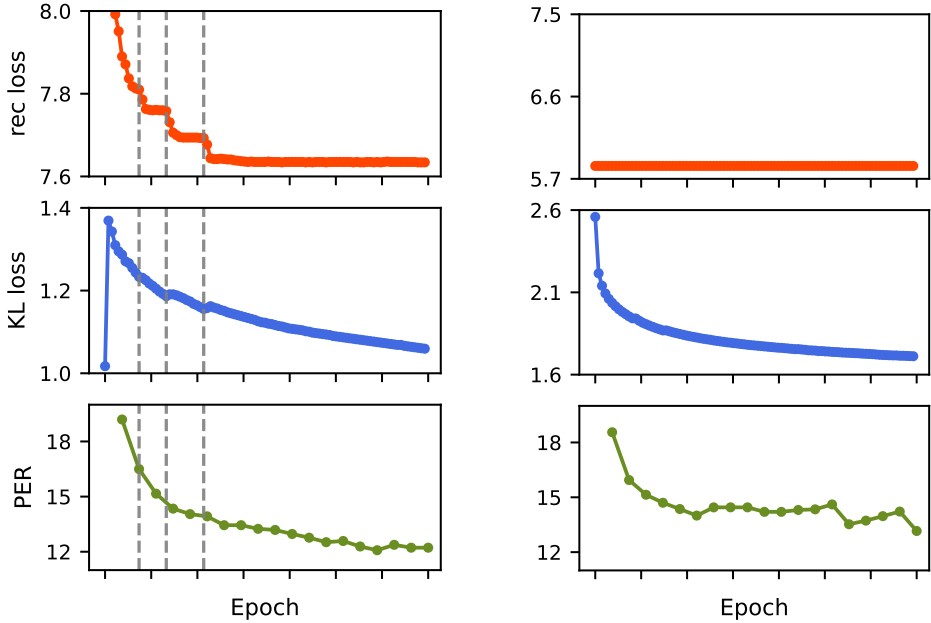

Figure 2: The learning dynamics of BASE MLM-VLB (left) and HuBERT (right) in terms of the rate and distortion. In HuBERT, the rate correspond to the cross entropy, the distortion is the k-means loss (fixed). The rate and distortion are calculated on the training data (LibriSpeech 960), while the PER curve is evaluated on dev93.

compared to the one with fewer bits. HuBERT representations, with the most bits, obtain the worst performance on ASR. This implies the model learns better representations when the distributions conditioned on different contexts are closer.

Table 3: We study the connection between the rate, distortion terms and ASR performance. A lower rate corresponds to better WERs on both testing sets.

|  | Rate | Distortion | dev93 | | eval92 | |
|---|---|---|---|---|---|---|
|  |  |  | CER | WER | CER | WER |
| HuBERT | 1.65 | 5.80 | 5.2 | 15.2 | 5.0 | 14.5 |
| MLM-VLB (k-means++) | 1.29 | 6.23 | 4.9 | 14.3 | 4.8 | 14.3 |
| MLM-VLB | 1.03 | 7.63 | 4.4 | 13.6 | 3.6 | 11.4 |

## 6 CONCLUSION

In this work, we have shown how the proposed variational objective on a predictive coding framework draws connections to previous work on self-supervised speech representation learning. In particular, our approach encompasses HuBERT, VQ-APC and links to other contrastive approaches. The proposed approach showcases strong performance on phonetic and speaker-related downstream tasks. Additionally, as indicated by the KL term, we have observed that a model learns better representations when the distributions on different parts of the signals are more closely aligned. The variational approach on predictive coding offers a new perspective on self-supervised learning objectives, encouraging the development of new training objectives under this framework.

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

## A   VARIATIONAL LOWER BOUND

We provide a derivation for variational lower bound of the proposed graphical model in Figure 1a. The goal is to approximate the posterior distributions with the variational distribution:

$$
\begin{aligned}
&D_{\mathrm{KL}}\big(q(z|x_B)\|p(z|x_A, x_B)\big) \\
&= \log p(x_B|x_A) + \mathbb{E}_{z\sim q}[\log q(z|x_B) - \log p(x_B|z) - \log p(z|x_A)] \\
&= \log p(x_B|x_A) + D_{\mathrm{KL}}\big(q(z|x_B)\|p(z|x_A)\big) - \mathbb{E}_{z\sim q}[\log p(x_B|z)],
\end{aligned} \tag{13}
$$

where we have assumed $p(x_B|x_A, z) = p(x_B|z)$ due to the independence assumption. Because the LHS of Equation 13 is always positive, we obtain the lower bound:

$$
\begin{aligned}
&-D_{\mathrm{KL}}\big(q(z|x_B)\|p(z|x_A)\big) + \mathbb{E}_{z\sim q}[\log p(x_B|z)] \\
&= \log p(x_B|x_A) - D_{\mathrm{KL}}\big(q(z|x_B)\|p(z|x_A, x_B)\big) \leq \log p(x_B|x_A)
\end{aligned} \tag{14}
$$

## B   PRE-TRAINING RECIPES

All models are built on the Transformers shown in Table 4. We set the number of centroids to 100 for k-means clustering to match a codebook size of 100. In wav2vec 2.0, the temperature $\kappa$ in equation 11 is 0.1, the temperature for Gumbel-Softmax (Jang et al., 2017) is annealed from 2 to a minimum of 0.5 by a decay rate of 0.999995.

| SMALL | parameters | | BASE | parameters |
|---|---|---|---|---|
| layers | 6 | | layers | 12 |
| attention heads | 4 | | attention heads | 6 |
| learning rate | $10^{-4}$ | | learning rate | $10^{-4}$ |
| learning rate schedule | constant | | learning rate schedule | constant |
| model dimension | 768 | | model dimension | 768 |
| inner dimension | 3072 | | inner dimension | 3072 |
| dropout | 0.1 | | dropout | 0.1 |
| codebook size | 100 | | codebook size | 100 |
| epochs | 100 | | epochs | 150 |
| batch size | 8 | | batch size | 16 |

Table 4: Transformer parameters.

Table 5: Results for ASR on WSJ compared to previous approaches. HuBERT 2nd iteration indicates the HuBERT model with another training iteration on refined cluster assignments. The refinement involves layer selection with phone labels.

| | Pre-train | dev93 | | eval92 | |
|---|---|---|---|---|---|
| | | CER | WER | CER | WER |
| BASE | | | | | |
| MLM-VLB | LS960 | 4.4 | 13.6 | 3.6 | 11.4 |
| wav2vec [†] | LS960 | 5.1 | 16.2 | 3.3 | 11.2 |
| vq-wav2vec [†] | | 7.0 | 20.4 | 4.5 | 14.7 |
| HuBERT 2nd iteration | LS960 + phone labels | 2.9 | 9.2 | 2.2 | 6.7 |

[†] Numbers taken from Baevski et al. (2020a).

# C  AUTOMATIC SPEECH RECOGNITION

We adopt a fixed scheduled sampling probability of 0.4 during training. We use Adam with learning rates of $10^{-4}$ for all s2s models. We employ a dropout rate of 0.2, and a label smoothing rate of 0.1 for regularization. For log Mel features and representations extracted from SMALL, we train s2s models for 200 epochs, and lower the learning rate with a factor of 0.1 for another 20 epochs. Regarding the representations extracted from BASE, we train s2s models for 100 epochs, and also lower the learning rate with a factor of 0.1 for another 20 epochs.

In addition to our own pre-trained models, we also compare the proposed approach to off-the-shelf models evaluated on WSJ (Schneider et al., 2019; Baevski et al., 2020a). We take the best layer of official released HuBERT BASE that has been trained for 2 iterations (Hsu et al., 2021), and evaluate the performance on our ASR system. Besides the architecture differences, the performance gap between HuBERT 2nd iteration and VLB can be explained with the addition iteration, in which they refine cluster assignments based on the best HuBERT layer from the first iteration. Note that the selection of layers involves phone labels. We do not consider another iteration since we are interested in the comparisons of loss functions by minimizing the architecture differences.

