# OpenReview forum: "A Discrete and Variational Approach to Speech Representation Learning"
_ICLR.cc/2024/Conference — ICLR 2024 Conference Withdrawn Submission_

### Official Review · Reviewer_G17g · 2023-10-31

**Soundness:** 3 good
**Presentation:** 2 fair
**Contribution:** 2 fair
**Rating:** 5
**Confidence:** 4

**Summary:**

The paper proposes a variational approximation of the speech representation learning problem that looks to generalize several previous works in the field and also provides advantages on the quality of the representation by imposing a direct relationship between the distribution of the latent representation given a known context (observed frames) and a variational distribution of the latent representation given the unknown context that must be reconstructed (masked or future frames).

The paper presents results on three standard downstream application tasks: Phone Classification, Speaker Verification, and Automatic Speech Recognition, and also evaluates the behavior of each component of the proposed ELBO, comparing it to the "equivalent" terms of HuBERT model. For the experimental phase, the paper uses simplified versions of previously proposed models used for comparison: wav2vec2.0 and HuBERT.

**Strengths:**

The proposed variational approximation tries to provide a better and principled formulation of the learning representation task, which contributes to a better understanding of the problem and presents a way to understand the relationships among the learning objectives of several of the existing solutions in the state-of-the-art. Moreover, it provides results that outperform simplified versions of two widely used models in the context of speech representation.

**Weaknesses:**

The formulation presented in the paper is not well-described; Figure 1b does not contribute to the understanding of the proposed approach and should be rebuilt entirely, and the learning process should also be explained in more detail. The use of simplified versions of benchmarking models limits the evidence of performance improvements presented in the paper.

**Questions:**

- How the proposed model guarantees the identifiability of distributions q(z_i|x_i) and p(z_i|x_\m) is unclear. Is the model train in fully unsupervised or (self-supervised) learning strategy or do the authors uses a force alligment to get the one-hot encoding vectors for all the experiments?

- The authors did not clarify their process to update the codebook; the whole learning process should be explained better.

- The authors did not perform any experiment to evaluate the effect of the codebook size, which was arbitrarily set to 100. According to previous results using VQ strategies for speech representation, that is a too-small value.

- Is there any reason that explains why the future prediction model outperforms the masked prediction training on speaker verification?

Minor things:

- The PER acronym is used before definition
-  There is an error in equiation (11) second row, las term should be p(z|x_A)

---

> ### Author Response · Authors · 2023-11-14
> **Author Response**
>
> We thank the reviewer for the valuable feedback.
>
> 1. There is a concern about the presentation of the submission. The
> original figure 1(b) is closer to the graphical rather than the
> feed-forward process that is more familiar in deep learning. We update
> a first draft of figure 1(b) to show the loss and the feed-forward process.
>
> > The formulation presented in the paper is not well-described; Figure 1b does not contribute to the understanding of the
> proposed approach and should be rebuilt entirely, and the learning process should also be explained in more detail.
>
> We have a draft version of [an updated figure 1(b)](https://i.imgur.com/XBfgSO4.png).
>
> > The authors did not clarify their process to update the codebook; the whole learning process should be explained better.
>
> In terms of the training process, the codebook V in equation in (5) and
> (6) are jointly optimized with gradient descent when minimizing equation
> (3).
>
> 2. There is a concern about our experimental methodology.
>
> > The use of simplified versions of benchmarking models limits the evidence of performance improvements presented in the p
> aper.
>
> Our focus is on the proposed loss function, and the benchmarking is
> exactly controlled under the same model architecture. The goal is more
> to show the connection among approaches, and less about getting the
> best numbers.
>
> 3. Miscellaneous questions
>
> > How the proposed model guarantees the identifiability of distributions q(z_i|x_i) and p(z_i|x_\m) is unclear.
>
> We are not sure about the identifiability mentioned here. We certainly
> make no guarantee that the auxiliary distribution q will be close to the
> posterior. We also want to caution about degeneracy. When the two sets
> A and B in equation (1) have overlaps, p and q can collaborate and lead
> to degeneracy. We will revise the paragraph before 2.1 to make this clear.
>
> > Is the model train in fully unsupervised or (self-supervised) learning strategy or do the authors uses a force alligment to get the one-hot encoding vectors for all the experiments?
>
> Our variational approach is unsupervised and self-supervised, and does
> not rely on forced alignments.
>
> > The authors did not perform any experiment to evaluate the effect of the codebook size, which was arbitrarily set to 100. According to previous results using VQ strategies for speech representation, that is a too-small value.
>
> We follow the setting in HuBERT. See Section IV. B. in Hsu et al. (2021) [1].
>
> > Is there any reason that explains why the future prediction model outperforms the masked prediction training on speaker verification?
>
> Our submission is the few to study speaker verification with causal
> self-attention, it is unclear at this point what the potential cause is.
>
> The key hyperparameters that decide what's learned in the representations
> are the time shift for future prediction and the masking probability
> and masking span in masked prediction. More research is needed to draw
> the connection between these hyperparameters and the learned representations.
>
> > Minor things:
>
> We thank the reviewer for catching the errors and will revise accordingly.
>
>
> [1] Hsu et al., HuBERT: Self-Supervised Speech Representation Learning by Masked Prediction of Hidden Units

---

### Official Review · Reviewer_9zbV · 2023-10-31

**Soundness:** 2 fair
**Presentation:** 2 fair
**Contribution:** 3 good
**Rating:** 3
**Confidence:** 5

**Summary:**

This paper proposes a new interpretation of self-supervised learning algorithms such as wav2vec 2.0 and quantized CPC in which a transformation of one part of the data is used to predict a quantized version of another part of the data.  The new formulation focuses on the quantizer, rather than focusing on the predictor: it uses a variational lower bound in which the log probability of the masked data given the unmasked data is bounded by the log probability of reconstruction from the codebook, minus the KL divergence between the quantizer distribution and the predictor distribution.

**Strengths:**

The theoretical argument is quite interesting.  The original wav2vec 1.0 paper included all of the components of the proposed approach, but in that paper, the codebook entropy was presented as sort of an ad-hoc method of avoiding mode collapse.  The KL divergence (information rate) suggested in this paper is a more principled way of understanding what wav2vec 2.0 is really calculating.

**Weaknesses:**

The theoretical argument is interesting, but the experiments are quite weak.  HuBERT and wav2vec 2.0 are crippled, and then the new representation is shown to outperform them.  Crippling the baselines might be forgivable if the crippling was irrelevant to the theoretical claims, but it is not.  HuBERT is crippled by not retraining the K-means codebook every few epochs, and wav2vec by removing the codebook entropy loss; these are directly relevant to the theoretical claims, and cause the experimental tests to be insufficient proof of the theoretical claims.

Against recommendations in the original HuBERT paper, this paper does not re-train HuBERT's codebook between epochs of transformer training. Figure 2 then shows that the proposed method achieves superior performance because it adapts the quantizer representation in a series of modes, which HuBERT cannot do because the authors chose not to allow it.  Indeed, re-training the K-means codebook in the manner recommended in the original HuBERT article would probably lead to a similar learning curve to VLB.

"As opposed to previous work in advocating codebook usage" -- The wording of this paragraph suggests that wav2vec increases a quantity while you decrease the same quantity, which is not true.  Your formulation measures D(q||p); diversity loss measures H(q).  Indeed, this is where the choice to remove H(q) from your wav2vec implementation is particularly troubling.  Wav2vec minimizes -H(q)-Eq[logp(z)], which is exactly D(q||p).  In other words, if you add back the entropy loss, wav2vec is already minimizing exactly the quantity proposed in this paper, and there should be no difference in performance between wav2vec and VLB.

Compared to those, this is a relatively minor point: One of the differences between future prediction and masked prediction is that, using future prediction, it's possible for each frame to serve two roles: to be predicted by its predecessor frames, while it is also a predictor of future frames.  Eq. (2) trivializes this by saying that the sum of all prediction log probabilities is less than or equal to the log probability of predicting the rest of the sequence from the first k frames.

There are a large number of grammar mistakes that.  Some of them, slowed my understanding of the paper somewhat: notable among these include the strange wording in the second line of the abstract, and the notational error in the second line of the equation in Appendix A.

p. 1

it is plausible if -> it is plausible that?  But why are you assuming that it is plausible?  I think, rather, you are proposing that this exists.

a information theoretic -> an information theoretic

and have a model -> and requiring a model

p. 4

$u_j$ is the j-th row of U -- I think you mean the j-th column.  Similarly v_j.

closet -> closest

self-supervise learning -> self-supervised learning

DeepCluter -> DeepCluster

p. 7

Table 1: The parameter count column for the BASE model contains the
string "LS960" rather than a parameter count.

p. 8

leanred -> learned

"representations achieve better downstream performance when fewer bits
are needed" -- I think this sentence belongs in the next paragraph; it
is not justified by any facts presented in this paragraph.

WERs degrades -> WER degrades

the model obtain -> the model obtains

Appendix A

Second line of Eq. (11): log p(z|XB) should be log p(z|XA).

**Questions:**

1. If you permit HuBERT to re-train its K-means codebook once every few epochs, does the resulting rate/distortion curve resemble the rate distortion curve of VLB?  What are the similarities and differences, and why?

2. If you permit wav2vec 2.0 to have its codebook entropy term, then is the resulting training criterion identical to VLB?  If not, why not?

---

> ### Author Response · Authors · 2023-11-14
> **Author Response Part 1**
>
> We thank the reviewer for the valuable and detailed feedback.
>
> There is a concern about our experimental methodology (quoted below)
> about the submission intentionally crippling the baselines. However, the
> reviewer seems to have misunderstood HuBERT and wav2vec. We will pinpoint
> where the confusion comes from and propose revision to our submission.
>
> > The theoretical argument is interesting, but the experiments are quite weak. HuBERT and wav2vec 2.0 are crippled, and then the new representation is shown to outperform them. Crippling the baselines might be forgivable if the crippling was irrelevant to the theoretical claims, but it is not.
>
>
> 1. Based on the quote below, the reviewer claims that HuBERT updates
> the k-means codebook during training.
>
> > HuBERT is crippled by not retraining the K-means codebook every few epochs ...
>
> > Against recommendations in the original HuBERT paper, this paper does not re-train HuBERT's codebook between epochs of transformer training. Figure 2 then shows that the proposed method achieves superior performance because it adapts the quantizer representation in a series of modes, which HuBERT cannot do because the authors chose not to allow it. Indeed, re-training the K-means codebook in the manner recommended in the original HuBERT article would probably lead to a similar learning curve to VLB.
>
> The above claim is technically incorrect, in the sense that the codebook
> in HuBERT is updated every gradient step, not every few epochs (See Hsu
> et al. (2021) for the term e_c in equation (3) [1] .). The targets (i.e.,
> that cluster assignment) to frames, however, are updated only after every
> iteration (See Hsu et al. Section IV. B.). The targets are updated with a
> new round of k-means, so a new codebook is produced every iteration, not
> every few epochs. K-means centroids across iterations are not compatible
> with each other. For example, each centroid in the first iteration has 39
> dimensions (for MFCCs), while each centroid in the second iteration has
> 768 dimensions (for HuBERT hidden vectors).
>
> Besides, iteration two and onwards in HuBERT are technically not
> unsupervised or self-supervised, because the selection of a HuBERT
> layer to produce targets relies on phone and cluster purity.
>
> We have stated explicitly in 5.1 that we do not do multiple iterations
> in our submission, because our approach can instantiate the HuBERT loss
> function for a particular iteration. In principle, it is possible to
> instantiate loss functions for multiple iterations, but that is not the
> goal of the paper. The goal of the paper is to show that it is possible
> to instantiate the HuBERT loss function with our proposed approach.
>
>
> 2. In terms of wav2vec 2.0, there is one concern about the diversity
> loss and about our experimental methodology.
>
> > "As opposed to previous work in advocating codebook usage" -- The wording of this paragraph suggests that wav2vec increases a quantity while you decrease the same quantity, which is not true. Your formulation measures D(q||p); diversity loss measures H(q). Indeed, this is where the choice to remove H(q) from your wav2vec implementation is particularly troubling. Wav2vec minimizes -H(q)-Eq[logp(z)], which is exactly D(q||p).
>
> We admit the wording issue for sending the wrong message. We agree
> that our entropy term serves the same purpose as the diversity loss in
> wav2vec 2.0.
>
> The point is that the diversity of codebook usage is encouraged and
> is part of the natural consequence of our variational bound, while the
> diversity loss in wav2vec 2.0 is given without any justification. (See
> Section 3.2 in Baevski et al. (2020) [2].) Another difference is that
> the diversity loss in wav2vec 2.0 is the entropy of an average within a mini-batch of utterances,
> while ours is computed per frame in Eq (3).
>
> > In other words, if you add back the entropy loss, wav2vec is already minimizing exactly the quantity proposed in this paper, and there should be no difference in performance between wav2vec and VLB.
>
> When instantiating wav2vec 2.0 in our approach, the entropy term has to
> be 0 because of q being one-hot. There is also no reconstruction term
> in wav2vec 2.0, which our approach has. We will revise section 4.3 to
> reflect the nuances.
>
> For the experimental methodology, the reviewer solely focuses on the
> cross entropy loss.
> > The original wav2vec 1.0 paper included all of the components of the proposed approach, but in that paper, the codebook entropy was presented as sort of an ad-hoc method of avoiding mode collapse.
> > Crippling the baselines might be forgivable if the crippling was irrelevant to the theoretical claims, but it is not. [...] and wav2vec by removing the codebook entropy loss ...
>
> The above claim ignores the fact that our loss function also includes
> a reconstruction term that is missing in wav2vec 2.0. Again, the entropy term has to be 0 because of the one-hot q.
>
> To avoid the confusion, we will revise section 4.3 and detail the
> difference between our approach and wav2vec 2.0.

---

> ### Author Response · Authors · 2023-11-14
> **Author Response Part 2**
>
> 3. There is a minor concern about how future prediction is formulated.
>
> > Eq. (2) trivializes this by saying that the sum of all prediction log probabilities is less than or equal to the log probability of predicting the rest of the sequence from the first k frames.
>
> We will rewrite the log probability of equation (2) as below to make clear the factorization of the autoregressive process:
> $L_{\text{AR}} \geq -\sum_{t=1}^{T-k}\log p(x_{t+k}|x_{\leq t})$
>
> 4. We thank the reviewer for the detailed feedback. We will revise the
> submission accordingly.
>
> > There are a large number of grammar mistakes that. Some of them, slowed my understanding of the paper somewhat: notable among these include the strange wording in the second line of the abstract, and the notational error in the second line of the equation in Appendix A.
>
> [1] Hsu et al., HuBERT: Self-Supervised Speech Representation Learning by Masked Prediction of Hidden Units
>
> [2] Baevski et al., wav2vec 2.0: A Framework for Self-Supervised Learning of Speech Representations

---

### Official Review · Reviewer_cFd9 · 2023-11-04

**Soundness:** 4 excellent
**Presentation:** 3 good
**Contribution:** 4 excellent
**Rating:** 8
**Confidence:** 4

**Summary:**

This paper proposes a variational learning framework for self-supervised learning. The authors studied the links between their framework and a few popular self-supervised learning approaches. More specifically, the authors show VQ-APC and HuBERT are all instances of the general framework they proposed.

The authors conduct experiments to demonstrate the advantage of their variational lower bound objective in terms of optimization. They observed sizable improvement in their experiments in phone classification, speaker verification and ASR. The authors also conduct analysis on the connection between learning dynamics and downstream ASR performance.

**Strengths:**

This is a very interesting work which may motivate a new angle for self-supervised learning. There are a couple of advantages:

1. This is the first work I’m aware of that tries to connect a few different self-supervised learning objectives and try to unify them under the same umbrella. Better understanding the connections of existing approaches, their connections, Pros and Cons are important.

2. The proposed VLB has benefits in terms of optimization.

3. The proposed approach provides an information theoretic len for analysis. Specifically, the authors analyzed the learning dynamics vs ASR performance which is motivated by the theoretical foundations laid out in Alemi et al. (2018) and Prokhorov et al. (2019).

4. The proposed approach achieves, if not state of the art, but sizable improvement on the baselines they have set up, which supports their claim on the optimization benefits.

**Weaknesses:**

I would not say these are really weak points, but may be bullet points the authors may pay attention to.

1. I think this is a very nice work, but maybe it is only 95% done presumably due to the ICLR submission deadline. I saw small typos at places. To name a few, in table 1, params should not be LS960, Sometimes, VLB was written as VLM, and some very minor writing typos.

2. The authors demonstrated the connection between their approach and VQ-APC and many more methods, but they only compared tow wav2vec2 and HuBERT. Also, the authors mostly only test one WSJ. To make the claim stronger, does it make sense to compare to more methods you have mentioned and tested on more downstream datasets?

3. Compared with wav2vec-2 and HuBERT, does the proposed framework have advantages or disadvantages in terms of GPU hours? This analysis could be interesting as the authors are proposing a general framework.

**Questions:**

1. In Table one and two, VLB-base archives even more significant improvement. Does this sound reasonable? My understanding is that, Table one and two are strong evidences on the optimization benefits of the variational framework; However, the baseline can be stronger with more tuning, better initialization, optimizer scheduler, and even more data; That is, the gap between the Hubert/wav2vec-2 and VLB in could much smaller than what is shown in this draft.

2. In Table three, are the rate and distortion calculated on dev93, eval92 or training data? Similar question to figure 2, is the PER curve on dev93 or eval92, or train?

---

> ### Author Response · Authors · 2023-11-14
> **Author Response**
>
> We thank the reviewer for the positive and valuable feedback. There are a few
> questions that we briefly answer below.
>
> > The authors demonstrated the connection between their approach and VQ-APC and many more methods, but they only compared to wav2vec2 and HuBERT.
>
> It is indeed interesting to compare to VQ-APC. Given the short timeline,
> we cannot guarantee to include VQ-APC during the rebuttal period, but
> will aim for the camera-ready version.
>
> > Compared with wav2vec-2 and HuBERT, does the proposed framework have advantages or disadvantages in terms of GPU hours?
>
> We do not observe a significant difference in GPU hours between HuBERT
> and the proposed method. We do find wav2vec-2 more computationally
> expensive due to the negative sampling step.
>
> > In Table one and two, VLB-base archives even more significant improvement. Does this sound reasonable? My understanding is that, Table one and two are strong evidences on the optimization benefits of the variational framework; However, the baseline can be stronger with more tuning, better initialization, optimizer scheduler, and even more data; That is, the gap between the Hubert/wav2vec-2 and VLB in could much smaller than what is shown in this draft.
>
> The gap might be smaller if we fine-tune on the downstream tasks instead
> of freezing the pre-trained models. However, all comparisons share almost the
> same hyperparameters, and we expect any improvement to the baseline to
> transfer to the proposed approach.
>
> > In Table three, are the rate and distortion calculated on dev93, eval92 or training data? Similar question to figure 2, is the PER curve on dev93 or eval92, or train?
>
> The rate and distortion are calculated on the training data (LibriSpeech
> 960 hr). The PER curve is evaluated on dev93. We will add more details
> to Table 3 and Figure 2.

---

### Official Review · Reviewer_jFUR · 2023-11-05

**Soundness:** 2 fair
**Presentation:** 3 good
**Contribution:** 2 fair
**Rating:** 3
**Confidence:** 3

**Summary:**

This paper proposed a new framework to unify causal and non-causal objective under a variational framework. Experimental results shows it's outperform Hubert and ablation also compared k-mean and on-the-fly learned codebook for the proposed VLB.

**Strengths:**

Unify causal and non-causal objective is a fundamental and important problem for audio representation.

**Weaknesses:**

(1) There are some analogy and connection to other model make no sense. For example, "The loss function becomes cross entropy if D contains all possible codes in V , and each code is uniquely sample", this is simply the difference of softmax and contrastive learning, I don't know what this rephrase means. I cannot see the proposed loss generalize anything to contrastive based approach.

(2) Based on (1), the proposed method is more like a unified version of w2v-bert [1] and best-rq [2], both of them using a mlm loss and learn the code on-the-fly without k-means.

(3) Experimental results are weak. No causal baseline been compared.

(4) The paper is unify causal (predictive) and non-causal (mask based), but none of such unification work been mentioned in the paper. Can the author survey and add it?

[1] W2v-BERT: Combining Contrastive Learning and Masked Language Modeling for Self-Supervised Speech Pre-Training
[2] Self-Supervised Learning with Random-Projection Quantizer for Speech Recognition

**Questions:**

Can the author explain more on the difference of proposed approach versus VQ-CPC?  Am I right conceptually it's replacing contrastive predictive coding with mlm loss?

---

> ### Author Response · Authors · 2023-11-14
> **Author Response**
>
> We thank the reviewer for the valuable feedback.
>
> 1. There is a confusion about how our approach links to contrastive
> methods, such as VQ-CPC. Our approach can instantiate VQ-CPC and wav2vec
> 2.0, but with an additional reconstruction term. We will make this clear
> in section 4.3.
>
> > (1) There are some analogy and connection to other model make no sense. For example, "The loss function becomes cross entropy if D contains all possible codes in V , and each code is uniquely sample", this is simply the difference of softmax and contrastive learning, I don't know what this rephrase means. I cannot see the proposed loss generalize anything to contrastive based approach.
>
> The contrastive loss is related to the cross-entropy term in the
> variational bound. Maximizing InfoNCE is analogous to maximizing the
> standard cross-entropy loss if negative samples always contain all
> possible values of codes (See Eq. 2 in Kong et al. [1]).
>
> > Can the author explain more on the difference of proposed approach versus VQ-CPC? Am I right conceptually it's replacing contrastive predictive coding with mlm loss?
>
> There is no reconstruction term in VQ-CPC, while there is one in our approach.
>
> 2. There is a concern about missing related work that does something
> similar to us.
>
> > (2) Based on (1), the proposed method is more like a unified version of w2v-bert [2] and best-rq [3], both of them using a mlm loss and learn the code on-the-fly without k-means.
>
> Indeed, w2v-BERT has both an MLM branch and a contrastive branch.
> (See Fig 2. in Chung et al. [2].) Our approach can instantiate MLM and
> a contrastive loss, but not both. The goal of our submission is to show
> that the two are special cases, not to have both at the same time.
>
> BEST-RQ does not update the codebook during training (See the abstract
> in Chiu et al. [3]), and can be seen as a simplified HuBERT. We will
> revise 4.2 to include a short discussion on this.
>
> 3. As quoted below, there is a concern about experimental methodology.
>
> > (3) Experimental results are weak. No causal baseline been compared.
>
> We do have results for a HuBERT trained with causal self-attention,
> and will update Table 2 accordingly. Note that our approach is already
> better than the causal baseline Baevski et al. (2020) [4] in Table 5, so the results are by no
> means weak.
>
> 4. We thank the reviewer for the detailed feedback. We will revise the
> submission accordingly.
>
> > The paper is unify causal (predictive) and non-causal (mask based), but none of such unification work been mentioned in the paper. Can the author survey and add it?
>
> The closest attempt is Kong et al. [1], and we will revise and point
> readers to related work therein.
>
> [1] Kong et al., A Mutual Information Maximization Perspective of Language Representation Learning\
> [2] Chung et al., W2v-BERT: Combining Contrastive Learning and Masked Language Modeling for Self-Supervised Speech Pre-Training\
> [3] Chiu et al., Self-Supervised Learning with Random-Projection Quantizer for Speech Recognition\
> [4] Baevski et al., vq-wav2vec: Self-Supervised Learning of Discrete Speech Representations

---

> > ### Comment · Reviewer_jFUR · 2023-11-22
> > **Thank you for the detailed reply.**
> >
> > Thank you for the explanation, that make the paper a lot clear to me.
> >
> > I want to clarify when I say weak experiments, I'm refer to WSJ is a barely used benchmark nowadays for ASR, and the reported results are weak. I understand the author argue that it's better than wav2vec 2 as Table 5, but please check some very simple baseline for WSJ: https://github.com/kaldi-asr/kaldi/blob/master/egs/wsj/s5/RESULTS
> >
> > I would keep my original score.

---

> ### Author Response · Authors · 2023-11-22
> **Re: Thank you for the detailed reply.**
>
> Thanks for your reply. We agree with that the absolute numbers are not better than Kaldi (and many other systems), but we respectfully disagree with the argument. We assume the differences among settings are clear, for example, that we are freezing the model to evaluate the representations, that we are using a relatively small end-to-end system, that we do not use a language model. Comparing to those absolute numbers misses the point of the paper and does not provide any evidence to support or to disprove any scientific argument.

---

### Official Review · Reviewer_sRTF · 2023-11-07

**Soundness:** 4 excellent
**Presentation:** 3 good
**Contribution:** 4 excellent
**Rating:** 8
**Confidence:** 4

**Summary:**

The authors present an innovative approach to self-supervised speech representation learning by adopting a variational perspective that unifies existing disparate methods under a predictive coding framework. By using a speech encoder that predicts certain data partitions from others, the system is able to learn predictive knowledge from the signal's context. This includes elements like phonetic details or speaker identity. The novelty lies in their proposition of a variational lower bound (VLB) on the log-likelihood for predicting context from input partitions, framing this process as a generative model with discrete latent variables.

This variational approach eliminates the need for an additional clustering step found in previous methods and provides a more efficient optimization strategy.

**Strengths:**

The strength of paper is that the method is not only aligned with but also extends the reach of other self supervised representation methods. Importantly, their VLB can draw parallels with contrastive objectives that aim to maximize mutual information.
Additionally, the authors explore the learning process through an information-theoretic lens, examining the interplay between KL loss (rate) and reconstruction loss (distortion) during training. They find that effective learning occurs in stages where these terms are balanced to achieve a stable latent distribution, leading to improved performance in downstream tasks when the KL divergence between disjoint contexts is minimized.

**Weaknesses:**

The authors should have more discussion and conclusion around those speaker verification downstream task, and discuss about why MLM-VLB performs better for phone recognition while causal-VLB performs better for speaker verification. More simulation and visualization of learned feature representations for an example sentence and compare it with other VQ based method would be beneficial and add more values to the work.

**Questions:**

The written English can be improved. There are few typos in different part of paper,  e.g. variey instead of variety in 2nd page.
Please revise and fix the problems.

---

> ### Author Response · Authors · 2023-11-14
> **Author Response**
>
> We thank the reviewer for the positive and valuable feedback.
> There is a question about the result on speaker verification.
>
> > More discussion and conclusion around those speaker verification downstream task, why does MLM-VLB perform better for phone recognition while causal-VLB performs better for speaker verification?
>
> Our submission is the few to study speaker verification with causal self-attention, it is unclear at this point what the potential cause is.
>
> The key hyperparameters that decide what's learned in the representations
> are the time shift for future prediction and the masking probability
> and masking span in masked prediction. More research is needed to draw
> the connection between these hyperparameters and the learned representations.
>
>
> > More simulation and visualization of learned feature representations for an example sentence and compare it with other VQ based method would be beneficial and add more values to the work.
>
> We thank the reviewer for the suggestion on adding more values to the work. We will add more visualization of the learned representations in the revision.

---

> > ### Comment · Reviewer_sRTF · 2023-11-22
> > **Adding figures of learned representation**
> >
> > Thanks! It would add more insight to the paper, and it would be great to do the same with the other approaches which your formulation could instantiate.
> > Please provide context information (phonetic boundaries) along your learned representations.

---

### Author Response · Authors · 2023-11-23
**Summary of revision**

We thank reviewer for the thorough and constructive feedback.

We are happy that the reviews find our paper "innovative" (reviewer zRTF),
addressing a "fundamental and important" problem (reviewer jFUR), "very
interesting" and motivating "a new angle for self-supervised learning"
(reviewer cFd9), with interesting "theoretical argument" (reviewer 9zbV),
providing "a better and principled forumation", and contributing "to a
better understanding" (reviewer G17g).

We have carefully addressed each point raised by the reviewers, and have
revised the submission.  We summarize the major changes below, and highlight them (red)
in the revision.

1. The connections to wav2vec 2.0 and VQ-CPC (reviewer jFUR and 9zbV):

There are three terms in the proposed variational bound.  The reviewers
have missed the reconstruction term and focused on the cross entropy
and entropy.  There is no reconstruction term in VQ-CPC and wav2vec 2.0,
differentiating our approach from the contrastive loss.  The InfoNCE loss
is related to cross entropy.  The diversity loss in wav2vec 2.0 is not
exactly the entropy term in our approach, but serves a similar purpose.
As expectation over q is approximated with Gumbel-Softmax, the entropy term is zero in practice.

Section 4.3 is now revised to make these points clear.

2. A misunderstanding of HuBERT and BEST-RQ (reviewer 9zbV and jFUR):

Reviewer 9zbV's claim about HuBERT training is technically incorrect
(see Author Response Part 1 for details).

Section 5.1 emphasizes that we do not do multiple iterations, as our
approach only aims to instantiate one iteration.  Section 4.2 mentions
the connection to BEST-RQ.

3. Additional causal baselines (reviewer jFUR and cFd9):

We have included HuBERT with causal attention on future prediction
with SMALL setup for linear probing tasks.  Given the short timeline,
we do not have the BASE results and VQ-APC at the moment, but will aim
for the camera-ready version.

|             | PER (dev93) | PER (eval92) | EER |
|:-----------:|:--------:|:----------:|:----------:|
|    Causal-VLB   | 18.5 |   18.8   | 15.6 |
| Causal-HuBERT |   21.5   |   19.5   | 15.1 |

4. The experimental methodology (reviewer 9zbV and jFUR):

We have emphasized in the rebuttal that goal is to hold the architecture
fixed while changing the loss functions.  This is not generally possible
unless clear theoretical connections are made as we do in the submission.

Our baselines are by no means weak. The goal is not to compete with
the best systems, but to establish fair comparison across loss functions
and to demonstrate the optimization benefit of jointly optimizating
the terms.

In addition, the coding perspective provides insights as to what
correlates with the downstream performance.

5. Wording and typos, figure 1b redesign (reviewer sRTF, cFd9, and 9zbV)

We have revised throughout the paper to improve the wording. We have also improved the feedforward plot in figure 1b.

---

### Meta-Review · Area_Chair_j8K5 · 2023-12-04

**Metareview:**

The authors advocate an innovative approach to self-supervised speech representation learning in the variational framework. The goal is to unify existing methods into a predictive coding framework. The theoretical argument presented in the paper is intriguing. The main weakness is the experimental validation, which is not convincing. There are also some concerns about the generalization capabilities of the proposed solution. In summary, the idea of having a single criterion is interesting, since it could help to derive something better than wav2vec or hubert. At present, the proposed work does not yet achieve that objective.

**Justification For Why Not Higher Score:**

Interesting idea that needs further validation.

**Justification For Why Not Lower Score:**

N/A

---

### Decision · Program_Chairs · 2024-01-16

Reject